# LUMEN-PRO: Automating Multi-Task Learning on Optical Neural Networks with Weight Sharing and Physical Rotation

## Abstract

With the demise of Moore's law, the demand for efficient deep neural network accelerators has surged. In addition, the democratization of AI encourages multi-task learning (MTL), demanding more parameters and processing time. To achieve highly energy-efficient MTL, Diffractive Optical Neural Networks (DONNs) have garnered attention due to extremely low energy and high computation speed. However, implementing MTL on DONNs requires manually reconfiguring and replacing specific layers, resulting in rebuilding and duplicating the physical systems. To overcome the challenges, we propose LUMEN-PRO, an automating MTL framework. Specifically, we first propose to automate MTL utilizing an arbitrary backbone DONN and a set of tasks, resulting in a high-accuracy multi-task DONN model with a small memory footprint that surpasses existing MTL methods. Secondly, we leverage the rotatability of the physical system, and replace task-specific layers with the rotation of the corresponding shared layers. This replacement eliminates the storage requirement of task-specific layers, thus further optimizing the memory footprint. LUMEN-PRO provides flexibility in identifying optimal sharing patterns across diverse datasets, facilitating the search for highly energy-efficient DONNs. Experimental results show that LUMEN-PRO provides up to $49.58\%$ higher accuracy and $4\times$ better cost efficiency than single-task and existing cutting-edge DONN approaches on different datasets. It achieves memory lower bound of multi-task learning, i.e., having the same memory storage as the single task model. Compared to technologies such as IBM TrueNorth and Nanophotonic, LUMEN-PRO achieves $10^5\times$ and $10\times$ speedup in throughput, and $5,969\times$ and $680\times$ energy efficiency gain, respectively.

## 1 Introduction

With the demise of Moore's law, the demand for efficient deep neural network accelerators has surged (Mack, 2015). To achieve real-time and ultra-low power DNN processing, advanced device/circuit technologies surpassing the Complementary metal–oxide–semiconductor (CMOS) technology are essential. As a representative, the Diffractive Optical Neural Network (DONN) has emerged to overcome the energy efficiency drawbacks associated with CMOS-based DNN systems (Caulfield & Dolev, 2010). The all-optical processing capabilities of DONNs are achieved by leveraging inherent physical phenomena, such as light diffraction and light signal phase modulation, occurring naturally at the speed of light. It offers (i) significantly less energy and thermal constraints, and more bandwidth compared to CMOS-based systems; and (ii) remarkable computational speed, transmitting information at the speed of light (Ríos et al., 2015; Shastri et al., 2021).

On the other hand, the increasing accessibility of AI encourages the concurrent execution of multiple interrelated tasks, i.e., more than one model simultaneously on a single resource-constrained device. This presents challenges due to the escalated computation, energy, and storage costs (Zhang & Yang, 2021). To tackle this, multi-task learning (MTL) provides a promising solution by facilitating joint learning of a task set and enabling parameter sharing to reduce costs. In general, MTL has high model complexity. For example, Kendall et al. (2018) shows their MTL model is $4.5\times$ slower at inference and requires $2.4\times$ more parameters than the single-task model for depth estimation when

using the same backbone network. The MTL model in Long et al. (2017) also shows $\sim 2\times$ more parameters than the single-task model and requires $\sim 4\times$ longer to train.

To realize highly energy efficient MTL, DONN technology appears to be a highly compelling and natural candidate, as evidenced by VanillaMT (Li et al., 2021a) and RubikONN (Li et al., 2023). Despite the success, the integration of MTL with DONN is still challenging due to the need (i) to rebuild and duplicate the physical hardware of the system, leading to energy disadvantage and cost inefficiency; (ii) of domain knowledge when designing resource-efficient MTL models, as substantial exploration efforts are needed to determine task-specific elements that shared across tasks.

Table 1: Comparison of different MTL methods.

| Methods | Multi-task | Automatic | Rotation |
|---|---|---|---|
| VanillaMT | ✓ | ✗ | ✗ |
| RubikONN | ✓ | ✗ | ✓ |
| (Ours) LUMEN-PRO | ✓ | ✓ | ✓ |

In this work, we propose an extremely energy-efficient automating multi-task learning framework on optical neural networks, LUMEN-PRO. LUMEN-PRO utilizes rotatability of the physical system and takes an arbitrary DONN backbone and a set of vision tasks as inputs, where the backbone defines the computation graph with layers functioning as shared operator nodes. The difference between our method and state-of-the-arts (SOTAs) are summarized in Table 1. Our contributions are summarized as follows:

- LUMEN-PRO automates the transformation of a user-provided DONN backbone into an operator-level supermodel, enabling gradient-based architecture search for efficient task sharing and cost optimization.

- LUMEN-PRO leverages the rotatability of the physical system to fine-tune the multi-task DONN architecture for resource efficiency. The task-specific layers are replaced by the physical rotation of the shared layers, therefore requiring zero memory footprint. Our method achieves a memory lower bound of multi-task learning, i.e., having the same memory storage as the single-task model.

- LUMEN-PRO can greatly reduce the cost and energy that required for MTL DONN applications, while still maintaining high prediction accuracy.

Experimental results show that LUMEN-PRO framework achieves up to $13.51\%$ higher accuracy and $4\times$ better cost efficiency than single-task and existing DONN baselines on MNIST family, with an improvement in prediction accuracy of up to $49.58\%$ on the CelebA. LUMEN-PRO also achieves up to $10^5\times$ and $10\times$ speedup in throughput, with $5,969\times$ and $680\times$ energy efficiency gain compared with IBM TrueNorth and Nanophotonic, respectively.

## 2 RELATED WORKS

### 2.1 DIFFRACTIVE OPTICAL NEURAL NETWORKS

Diffractive Optical Neural Networks (DONNs) is an optical system where the information encoding and the computation are realized by the manipulation of the light signal, which features with high energy efficiency, high computational speed and easy parallelism (Shen et al., 2017; Lin et al., 2018; Mengu et al., 2019; Feldmann et al., 2019; Li et al., 2021b; 2022). The DONN system is composed by stacking diffractive layers in sequence as shown in Figure 1. The input information is encoded with the coherent light signal on its optical characteristics, e.g., its intensity, amplitude, or phase. The diffractive layers are arrays embedding the phase modulations trained w.r.t the ML task for manipulating and encoding information on the light signal. The connection between layers is realized by the light diffraction when the light signal propagates between layers. At the end of the DONN system, a detector is employed to capture the light intensity pattern for the result readout and the analog-to-digital conversion. Note that the optical manipulation happens by nature with light propagation and modulation and the diffractive layers are implemented with passive optical devices without extra energy needed for functionality, resulting in ultrahigh power efficiency and computational speed. Once the training of a DONN system is completed on the digital computation platform, the trained DONN is deployed on the optical platform with non-configurable fabricated phase masks such as 3D printed phase masks, as diffractive layers for all-optical inference. Thus,

DONNs lack reconfigurability for the weight parameters, which will bring significant energy and system cost overhead in practical application scenarios, especially for MTL.

## 2.2 MULTI-TASK LEARNING

There are several research trends addressing the multi-task learning efficiency: (1) common features extraction with feature transformation (Weinberger et al., 2009; Yim et al., 2015; Chu et al., 2015; Zhang et al., 2014) and feature selection approaches (Wang et al., 2009; Ahmed et al., 2012), (2) low-rank methods for weight parameter approximation and sharing (Xu et al., 2016; Cheng et al., 2011), (3) clustering tasks based on task similarity (An et al., 2008; Zhang & Yeung, 2010; Almaev et al., 2015), (4) simultaneous learning of parameters and pairwise task relations (Chapelle et al., 2010; Liu et al., 2016; Widmer et al., 2010), and (5) decomposition approaches using multi-level parameters to model complex task structures (Hong et al., 2013; Yan et al., 2013; Wan et al., 2012).

## 3 DONN PRELIMINARY

DONN system is designed with three major components (Figure 1): (1) laser source encoding the input images, (2) diffractive layers encoding trainable phase modulation, and (3) detectors capturing the output of the forward propagation. The input image is first encoded with the laser source. The information-encoded light signal is diffracted in the free space between diffractive layers and modulated via phase modulation at each layer. Finally,

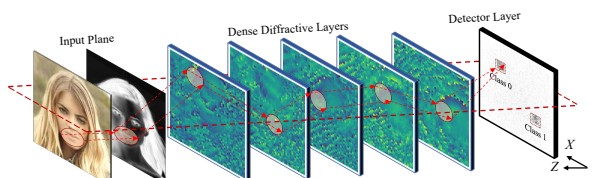

Figure 1: A Five-layer DONN for classification.

the diffraction pattern after light propagation w.r.t light intensity distribution will be captured at the detector plane for predictions.

First, the input information (e.g., an image) is encoded with the coherent light signal from the laser source, and the information-encoded wavefunction is $f^0(x_0, y_0)$. The wavefunction after light diffraction from the input plane to the first diffractive layer over diffraction distance $z$ can be seen as the summation of the outputs at the input plane, i.e.,

$$f^1(x, y) = \iint f^0(x_0, y_0) h(x - x_0, y - y_0, z) dx_0 dy_0 \tag{1}$$

where $(x, y)$ is the coordinate on the receiver plane, i.e., the first diffractive layer, $h$ is the impulse response function of free space. Here we use Fresnel approximation, thus the impulse response function $h$ is as Equation 2, where $i = \sqrt{-1}$, $\lambda$ is the wavelength of the laser source, $k = 2\pi/\lambda$ is free-space wavenumber.

$$h(x, y, z) = \frac{\exp(ikz)}{i\lambda z} \exp\{\frac{ik}{2z}(x^2 + y^2)\} \tag{2}$$

Equation 1 can be calculated with spectral algorithm, where we employ Fast Fourier Transform (FFT) for fast and differentiable computation, i.e., $U^1(\alpha, \beta) = U^0(\alpha, \beta)H(\alpha, \beta, z)$, where $U$ and $H$ are the Fourier transformation of $f$ and $h$.

After light diffraction, the wavefunction resulting $U^1(\alpha, \beta)$ is first transformed to time domain with inverse FFT (iFFT). Then the phase modulation $W(x, y)$ provided by the diffractive layer is applied to the light wavefunction in time domain by matrix multiplication, i.e., $f^2(x, y) = $ iFFT$(U^1(\alpha, \beta)) \times W_1(x, y)$, where $W_1(x, y)$ is the phase modulation in the first diffractive layer, $f^2(x, y)$ is the input light wavefunction for the light diffraction between the first diffractive layer and the second diffractive layer.

The computation module with one computation round of light diffraction and phase modulation at one diffractive layer is named **DiffMod**, i.e.,

$$\text{DiffMod}(f(x, y), W) = L(f(x, y), z) \times W(x, y) \tag{3}$$

where $f(x, y)$ is the input wavefunction, $W(x, y)$ is the phase modulation, $L(f(x, y), z)$ is the wavefunction after light diffraction over a constant distance $z$ in time domain, i.e., iFFT$(U(\alpha, \beta))$.

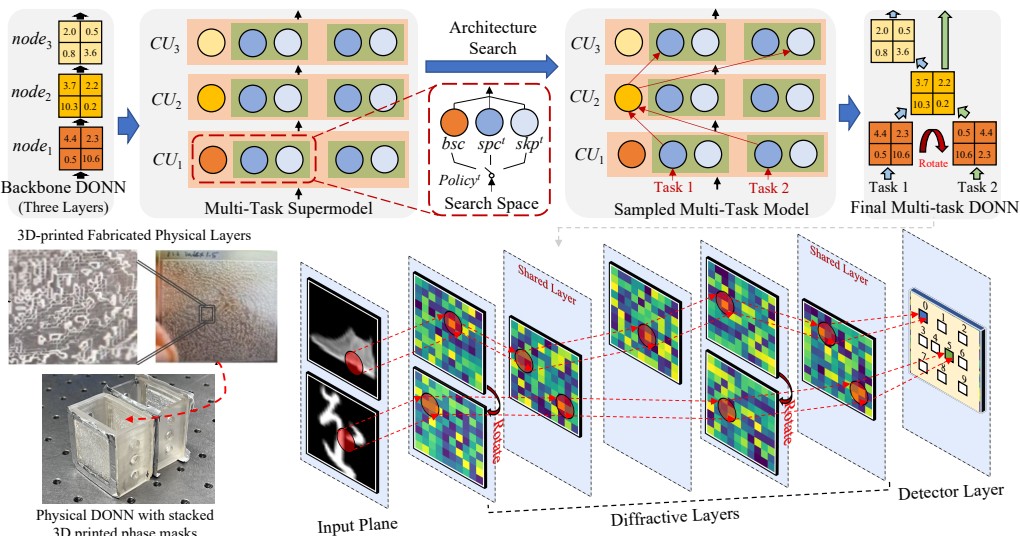

Figure 2: Overview of the proposed LUMEN-PRO framework and DONNs system.

The forward function for a multiple-diffractive-layer-constructed DONN system is computed iteratively for the stacked diffractive layers. For example, for the 5-layer system shown in Figure 1, the forward function can be expressed as,

$$I(f^0(x, y), W) = \text{DiffMod}(\text{DiffMod}(\text{DiffMod}(\text{DiffMod}(\text{DiffMod}$$
$$(f^0(x, y), W_1(x, y)), W_2(x, y)), W_3(x, y)), W_4(x, y)), W_5(x, y)) \tag{4}$$

where $f^0(x, y)$ is the input wavefunction to the system and $W_{1-5}$ is phase modulation provided at each diffractive layer.

The final diffraction pattern w.r.t the light intensity $I$ as denoted in Equation 4 is projected onto the plane of the detector. By defining the coordinates of the detector region across the entire detector plane for each class according to the user's specifications, it becomes feasible to devise diverse detector patterns for various tasks. As an illustration, in the case of MNIST datasets, the output plane is partitioned into **ten** distinct detector regions to emulate the outcomes of conventional neural networks that predict **ten** classes. The classification result is determined by employing the argmax function on the sums of intensities from the ten detector regions. For instance in Figure 2, by examining the label indices of the ten detector regions corresponding to the image "boots", the highest energy is observed in the first region of the first row. Consequently, the predicted class is "0". By utilizing the one-hot encoded representation of the ground truth class denoted as $t$, the loss function $L$ can be obtained through the utilization of **MSELoss**, i.e., $L = \parallel \text{Softmax}(I) - t \parallel_2$. Thus, the whole system is designed to be differentiable and compatible with conventional automatic differential engines.

## 4 LUMEN-PRO FRAMEWORK

We aim to develop a precise, cost-efficient automating multi-task learning DONN system by sharing parameters across tasks. Diffractive layers in DONN systems are typically 3D printed and feature permanent phase parameters once fabricated. However, their square shape allows for relocating and rotating the layers, enabling weight alterations and modification of the DONN system's forward function (Lin et al., 2018). This rotation changes light modulation patterns and enhances the performance and computational efficiency of the MTL DONN system. Figure 2 provides an overview of our LUMEN-PRO framework.

### 4.1 AUTOMATING MULTI-TASK LEARNING FRAMEWORK

The multi-task supermodel is generated from a backbone DONN as shown in Figure 2, encoding the search space for all tasks. A gradient-based architecture search algorithm is employed to find the

optimal architecture, ensuring accuracy and compactness. Task-specific aspects are then addressed, where task-specific copies replace weights with rotated weights from the shared operator of the backbone model. This approach tailors the model for each task without increasing energy and cost compared to a single-task model.

### 4.1.1 SUPERMODEL AND SEARCH SPACE

Our framework views the DONN backbone model as a computational graph, with diffractive layers as operators that serve as sharing units (Figure 3). Sharing options include using (1) the basic shared operator ($bsc$), (2) a task-specific copy of the basic shared operator ($spc^t$), or (3) a skip operator ($skp^t$). The multi-task DONN supermodel is represented by a computational graph using Computation Units ($CUs$) as layers, preserving the backbone model's topology. These $CUs$ include the necessary components: the basic shared operator, task-specific copies, skip operators, and trainable policy variables ($P^t$) for task execution and sharing patterns.

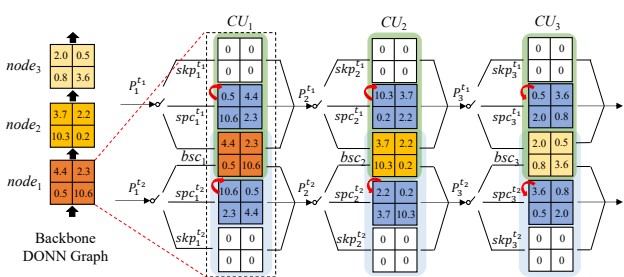

Figure 3: Computational graph of LUMEN-PRO.

### 4.1.2 GRADIENT-BASED ARCHITECTURE SEARCH

Our aim is to discover the most effective sharing policy for a multi-task supermodel that results in the top-performing outcome across all tasks. To effectively explore the search space and identify the ideal sharing policy for a multi-task DONN supermodel, a gradient-based architecture search algorithm is employed (Zhang et al., 2022). This algorithm optimizes the sharing policy and multi-task DONN model parameters simultaneously using standard back-propagation. The non-differentiability and discrete nature of policy variables are handled using the Gumbel-Softmax Approximation (Jang et al., 2016) and soft differentiable policy as Equation 5.

$$P'(k) = \frac{\exp\left(\left(G_k + \log\left(\pi_k\right)\right)/\tau\right)}{\sum_{k \in \{0,1,2\}} \exp\left(\left(G_k + \log\left(\pi_k\right)\right)/\tau\right)} \tag{5}$$

Here, $P$ is the policy variable; $k$ represents the three operator options that 0 is the backbone basic shared operator $bsc$ is chosen for the task, 1 is the rotated task-specific copy $spc^t$ is adopted, and 2 is the skip operator $skp^t$ is selected; $G_k \sim Gumbel(0,1)$. Once the distribution $\pi$ is learned, we sample the discrete task-specific policy $P$, which determines the operator to execute in each $CU$ for each task. Using this policy, we construct the multi-task DONN architecture, ensuring better performance among all the tasks.

To further optimize the energy and cost overhead, sharing operators across tasks are more encouraged in the multi-task DONN model. Denote the probability of selecting the basic shared operator, the task-specific copy, and the skip operator as $P'^{t_m}_n(0)$, $P'^{t_m}_n(1)$, and $P'^{t_m}_n(2)$ for the $m$-th task in the $n$-th $CU$, a policy regularization term $L_{reg}$ (Dugas et al., 2000) is added to the loss function as Equation 6. Here, $T$ is the total number of tasks, and $N$ is the total number of diffractive layers in the backbone model.

$$L_{reg} = \sum_{m \leq T} \sum_{n \leq N} \frac{N-n}{N} \left\{ \ln\left(1 + e^{P'^{t_m}_n(1) - P'^{t_m}_n(0)}\right) + \ln\left(1 + e^{P'^{t_m}_n(2) - P'^{t_m}_n(0)}\right) \right\} \tag{6}$$

In this case, the final loss function can be written as Equation 7. Here, $L_m$ refers to the loss for each task as shown in Section 3, $\alpha_m$ and $\alpha_{reg}$ are regularization factors.

$$\mathcal{L} = \sum_{m \leq T} \alpha_m \cdot L_m + \alpha_{reg} \cdot L_{reg} \tag{7}$$

### 4.2 ROTATION ALGORITHM

We use a three-stage training pipeline for the multi-task DONN model. In the first stage (Figure 2), pre-training is conducted jointly on all tasks to obtain a well-initialized multi-task DONN supermodel. The output of each *CU* for each task is the average of the backbone shared operator, task-specific copy, and skip connection, ensuring parameter warming. The second stage is policy-training, optimizing the sharing policy and model parameters iteratively. Once the policy distribution parameters converge, a sharing policy is sampled to generate the multi-task DONN model. In the post-training stage, the sharing policy is fixed, and

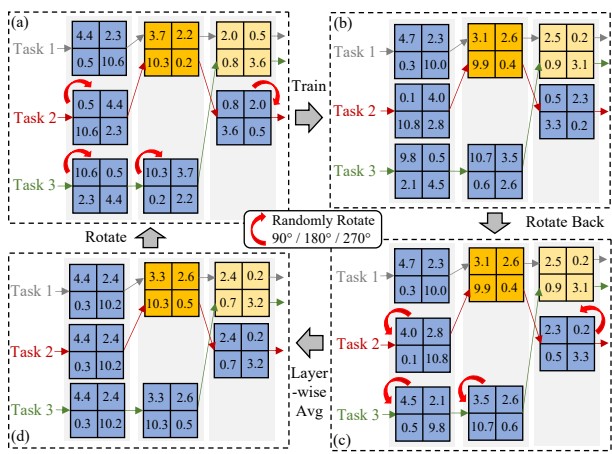

Figure 4: Rotation Process in LUMEN-PRO.

model parameters are trained from scratch. We use the rotation training algorithm to leverage the inherent physical rotation properties of DONN systems for MTL with minimal overhead.

After the sampling stage, the structure of the multi-task model is finalized, and operator selection for each node in each task is determined. As in Algorithm 1, during the post-training phase, two models are initialized: one for aggregation and another as a virtual model to temporarily store updates for specific rotation patterns and tasks. The virtual model is re-initialized with either the initial weight parameters or the parameters optimized in the previous iteration. As in Figure 4(a), during each training iteration, nodes in certain tasks may choose the shared backbone operator or the task-specific copy in a specific layer of the multi-task DONN model. The weights in the task-specific copies are then replaced with varying degrees of rotation using the weights from the backbone shared operator (lines 3-5). After rotations and substitutions, the parameters in the virtual model are updated (line 6). After one training iteration as in Figure 4(b), all substituted weights in the virtual

---

**Algorithm 1:** LUMEN-PRO Rotation Algorithm

**Input:** A MTL DONN $M$ with $N$ layers, $T$ tasks
**Initialization:** A copy of $M$ as $M_{agg}$

1   Extract the policy set from $M$ as $\{\mathbf{P}\}$
2   **while** $i \leq training\_iterations$ **do**
3     **for** $n \leq N$ **do**
4       **if** $spc_n^t$ $in$ $\{\mathbf{P}_n\}$ **then**
5         Substitute the weights of $t$ in $M$ with the weights of $bsc_n$ in $M_{agg}$ after applying a rotation of $\alpha_n^t$ degrees.
6     Train and update weights in $M$.
7     **for** $n \leq N$ **do**
8       Apply a rotation of $4 - \alpha_n^t$ degrees to substituted weights in $M$.
9       Compute the average of the weights of all tasks in layer $n$ in $M$, denote as $\mathbf{W}_n^{tmp}$.
10      **for** $t \leq T$ **do**
11        Replace weight of $t$ in $M_{agg}$ with $\mathbf{W}_n^{tmp}$.
12        **if** $policy_n^t$ $is$ $spc_n^t$ **then**
13          Substitute the weights of $t$ in $M$ with $\alpha_n^t$-degrees rotated $\mathbf{W}_n^{tmp}$.
14     Evaluation on $M$.

**Output:** Well-trained MTL DONN $M$ w/ rotated weights.

---

model are reverse-rotated back to their initial position (line 7-8) as in Figure 4(c). Aggregation is then performed by averaging weights of nodes across tasks in the same layer (lines 9-11), and new weights are copied to re-initialize the aggregation model (line 12-13) as in Figure 4(d).

## 5 EXPERIMENTS

### 5.1 SYSTEM PARAMETERS AND TRAINING SETUP

**Dataset and Evaluation Metrics** We evaluate the performance of LUMEN-PRO on two popular multi-task learning datasets. The first is MNIST family, which consists of four public image classification datasets: MNIST-10 (MNIST) (LeCun, 1998), Fashion-MNIST (FMNIST) (Xiao et al., 2017), Kuzushiji-MNIST (KMNIST) (Clanuwat et al., 2018), and Extension-MNIST-Letters (EM-

NIST) (Cohen et al., 2017). For EMNIST, we customize the dataset by selecting the first ten classes "A-J". The second used is CelebFaces Attributes Dataset (CelebA) (Liu et al., 2015). We choose four attributes with relatively more balanced labels, namely *Smiling, Mouth_Slightly_Open, Male, and Attractive*. We transform all images to grayscale to ensure compatibility with DONN physical system. $F_1$-score and accuracy are used as evaluation metrics for CelebA.

**DONN System Parameters and Training Setup**  We utilize a system configured with five diffractive layers, each of dimension $200 \times 200$, hence both the layer and the detector regions share these dimensions. The system is configured in $532nm$ laser wavelength (green laser). Original input images in the evaluated datasets are interpolated to $200 \times 200$ to align with our optical system and encoded. We maintain a uniform physical distance of $27.94cm$ among layers, between the first layer and the source, and from the final layer to the detector. The distinct detector regions, corresponding to the number of classes, are uniformly situated on the detector plane, each sized $20 \times 20$. The aggregate intensity of the detector regions equates will return a vector in `float32` type. The final prediction results are computed using `argmax`. During the training process, we use learning rate as $0.1$ under Adam optimizer, batch-size being $200$. *StepLR* scheduler is adoped with step-size being $6000$. All the implementations are constructed using `PyTorch v1.8.1`. Experiments are conducted on a Nvidia Quadro RTX6000.

## 5.2 EVALUATION RESULT

### 5.2.1 ACCURACY COMPARISON

We compare LUMEN-PRO with four baselines: (1) a single-task baseline with independent models for each task, (2) *BaselineMT* (Lin et al., 2018), a straightforward multi-task approach using a fixed single-task DONN architecture by merging datasets, (3) *VanillaMT* (Li et al., 2021a), a specific MTL DONN method with shared backbone and separate diffractive layers at the output stage, and (4) RubikONN (Li et al., 2023), a state-of-the-art MTL

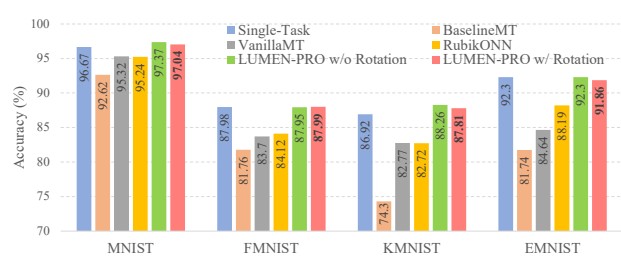

Figure 5: Comparison with baselines on MNIST Family.

DONN method with weight aggregation and rotation but manual architecture design. We select the best combination of rotated layers and rotation angles for RubikONN. To ensure fair comparisons, we use the same backbone DONN model across all baselines and LUMEN-PRO.

**MNIST Family**  Figure 5 compares the accuracy of LUMEN-PRO with and without rotation to baseline methods on four MNIST family datasets. We use MNIST family to align with the baselines. Results show that LUMEN-PRO outperforms all MTL baselines, especially BaselineMT, with improvements of $4.42\%$, $6.23\%$, $13.51\%$, and $10.12\%$ in MNIST, FMNIST, KMNIST, and EMNIST datasets, respectively. LUMEN-PRO with rotation achieves significant accuracy gains of $1.8\%$, $3.87\%$, $5.09\%$, and $3.67\%$ compared to RubikONN. Compared to the single-task baselines, LUMEN-PRO with rotation demonstrates accuracy improvements of $0.37\%$ (MNIST) and $0.89\%$ (KMNIST). Notably, there is minimal difference in performance between LUMEN-PRO with and without rotation, and rotation does not incur additional memory usage on the physical system for the multi-task DONN model.

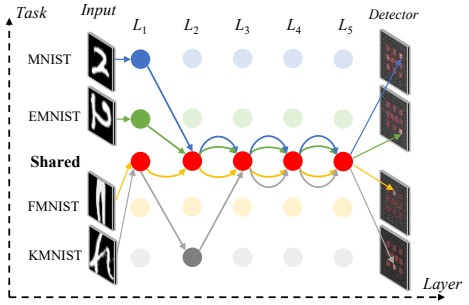

Figure 6: Policy Visualization for MNIST Family. Weights on non-red nodes derived via rotational transformation based on red (shared) node weights. Semi-transparent nodes mean the operators are not selected.

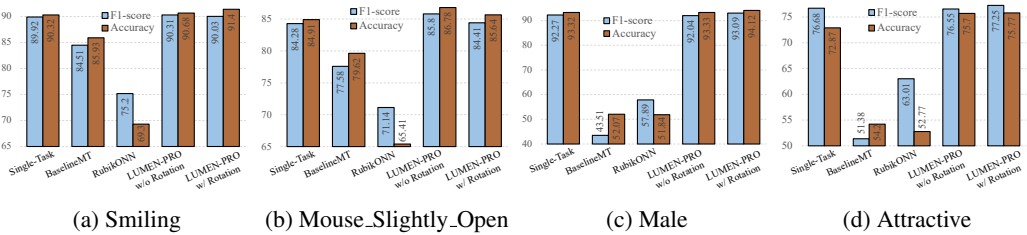

| (a) Smiling | (b) Mouse_Slightly_Open | (c) Male | (d) Attractive |

Figure 7: Comparison of single-task and MTL baselines with LUMEN-PRO on CelebA dataset.

We analyze the learned sharing policies of the multi-task DONN architecture. Figure 6 displays the sampled feature sharing pattern for the four datasets. Earlier diffractive layers tend to have dataset-specific weight selection, while operator sharing primarily occurs in later layers. FMNIST consistently shares across all five diffractive layers. These results contrast with the architecture design from RubikONN, where the first three diffractive layers are shared across tasks and the fourth and fifth layers are task-specific and rotated from the backbone operators at pre-designed angles.

**CelebA** Figure 7 compares LUMEN-PRO to baselines in a four-task MTL with CelebA attributes. LUMEN-PRO outperforms the single-task model in all tasks, indicating task correlations that enable our MTL DONN model to leverage shared information and learn common features, improving overall performance. With rotation, LUMEN-PRO outperforms BaselineMT by up to $49.58\%$ and RubikONN by up to $35.2\%$. Unlike BaselineMT and RubikONN, our LUMEN-PRO framework automatically explores and tailors architectures for data and task specifics, enhancing correlation and dissimilarity capturing for improved performance across tasks.

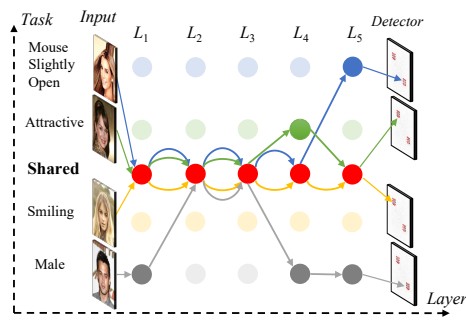

Figure 8: Policy Visualization for CelebA.

Figure 8 provides insights into the architecture of the multi-task DONN model designed for CelebA. In the initial layers, shared operators are utilized, whereas rotated task-specific operators are employed starting from the fourth layer onwards. Our approach consistently outperforms RubikONN-designed models, accommodating task heterogeneity effectively. The architecture for CelebA differs significantly from the MNIST-based dataset, demonstrating the flexibility of our automated feature in LUMEN-PRO for diverse tasks and datasets.

### 5.2.2 Accuracy-Cost Comparison

In this part, we compare the model efficiency of LUMEN-PRO in terms of system cost and accuracy to MTL baselines and single-task models. We use the "Accuracy-Cost evaluation metric" as in Equation 8 for this evaluation (Li et al., 2023).

$$E_{acc-cost} = \frac{\text{Accuracy}_{single-task}}{\text{Accuracy}_{MTL}} \times \frac{\text{F-Cost}_{MTL}}{\text{F-Cost}_{single-task}} \quad (8)$$

Here, F-cost represents the fabrication cost for the diffractive layers and the detectors in the hardware implementation, where the layer fabrication cost is $\sim \$100$ and detector cost is $\$1,500 - \$10,000$. We normalize the cost of $\$100$ as unit 1, thus, the layer cost for a 5-layer ONN is 5 and one detector cost is 10 for the cost efficiency estimation.

Table 2 compares the accuracy and model efficiency of LUMEN-PRO with single-task models and MTL baselines. The model efficiency of single-task models is normalized to a unit value of 1, which is used to calculate the improvement achieved by the MTL method using Equation 8. The sharing ratio indicates the proportion of shared layers among all four tasks. Both VanillaMT and RubikONN share the first three layers and diverge at the last two, with RubikONN reusing the rotation layers. As shown, LUMEN-PRO with rotation surpasses the single-task model and VanillaMT with $4\times$ and

Table 2: Accuracy-Cost trade-off comparsion of MNIST family: MNIST (Mnst), FMNIST (Fmst), KMNIST (Kmnist), EMNIST (Emst). LUMEN-PRO follows the model structure shown in Figure 6.

| | Single-task | | | | VanillaMT | | | | RubikONN | | | | LUMEN-PRO w/o Rotation | | | | LUMEN-PRO w/ Rotation | | | |
|---|---|---|---|---|---|---|---|---|---|---|---|---|---|---|---|---|---|---|---|---|
| Task | Mnst | Fmst | Kmst | Emst | Mnst | Fmst | Kmst | Emst | Mnst | Fmst | Kmst | Emst | Mnst | Fmst | Kmst | Emst | Mnst | Fmst | Kmst | Emst |
| Accuracy (%) | 96.4 | 86.5 | 86.1 | 90.9 | 95.3 | 83.7 | 82.8 | 84.6 | 95.2 | 84.1 | 82.7 | 88.2 | 97.4 | 87.9 | 88.3 | 92.3 | 97.0 | 88.0 | 87.8 | 91.7 |
| Layer Cost (Norm.) | 5 | 5 | 5 | 5 | 11 | | | | 5 | | | | 8 | | | | 5 | | | |
| Detector Cost (Norm.) | 10 | 10 | 10 | 10 | 20 | | | | 10 | | | | 10 | | | | 10 | | | |
| Model Eff. (Norm.) | 1 | 1 | 1 | 1 | 1.8× | 1.76× | 1.75× | 1.69× | 3.95× | 3.89× | 3.84× | 3.88× | 2.48× | 2.46× | 2.44× | 2.46× | 4.03× | 4.07× | 4.08× | 4.04× |
| Sharing Ratio | – | | | | 12/20 | | | | 16/20 | | | | 16/20 | | | | 17/20 | | | |

Table 3: Comparison on accuracy, performance, energy efficiency of LUMEN-PRO and baselines

| | FINN | TrueNorth | FORMS | ISAAC | DaDianNao | Holylight | LUMEN-PRO |
|---|---|---|---|---|---|---|---|
| Technology | FPGA | ASIC | ReRAM | ReRAM | ASIC | Photonic | Free-space Optics |
| Network Type | BNN | SNN | CNN | CNN | CNN | CNN | DONN |
| Accuracy (%) | 98.4 | 95 | 99.17 | 99.1 | 99.18 | 98.9 | 97 |
| Throughput (kFPS) | 1561 | 1.0 | 500 | $1 \times 10^{3*}$ | 100* | $1 \times 10^{4*}$ | $1 \times 10^5$ |
| Power (Watt) | 22.6 | 0.06 | 66.36 | 65.8 | 20.1 | 68.3 | 1.005 |
| Energy Eff. (kFPS/W) | 69.07 | 16.67 | 7.53 | 15.2 | 4.98 | 146.4 | $9.95 \times 10^4$ |

* Note: Our retrieved numbers from the histograms in Liu et al. (2019), to the best of our knowledge.

$2\times$ model efficiency improvements, respectively. It has a $50\%$ higher sharing ratio than VanillaMT, reducing fabrication costs during deployment. Compared to RubikONN, LUMEN-PRO consistently achieves higher model efficiency and accuracy across all tasks, demonstrating its effectiveness in diverse datasets and real-world applications.

Comparing our LUMEN-PRO with and without rotation algorithm, we find that without rotation, LUMEN-PRO achieves higher accuracy but lacks the cost-saving benefits of reusing middle layers. This highlights the importance of balancing accuracy and cost in real-world deployment. By incorporating the rotation algorithm, LUMEN-PRO enables high-performing structures that excel in both accuracy and model efficiency, improving practical applicability.

**Energy Efficiency Comparison** In Table 3, we provide the comparison results on accuracy, throughput, power, and energy efficiency on MNIST. We select the state-of-the-art extremely energy efficiency implementation from other technologies (e.g., FPGA, ASIC, ReRAM) as baselines, including FINN (Umuroglu et al., 2017) (binary neral network (BNN)), IBM TrueNorth (Esser et al., 2015) (spiking neural network (SNN)), FORMS (Yuan et al., 2021), ISAAC (Shafiee et al., 2016), DaDianNao (Chen et al., 2014), and Holylight (Liu et al., 2019) (photonic CNN). We observe that compared to the reference FPGA-based implementation at a similar accuracy level, we achieve $64\times$ speedup in throughput, while the energy efficiency gain is $1,441\times$. In comparison to ReRAM-based and ASIC-based implementations, the energy efficiency gain is at least $6,546\times$ and $5,969\times$ respectively. And when compared to photonic CNN, there is $10\times$ throughput speedup and $680\times$ energy efficiency gain.

## 6 CONCLUSIONS

In this paper, we propose LUMEN-PRO, an automating multi-task learning optical neural network framework that optimizes MTL DONN using physical principles. LUMEN-PRO converts a user-provided DONN backbone into a supermodel with operator-level granularity, enabling gradient-based architecture search for optimal possible sharing patterns across multiple tasks, enhancing cost efficiency. LUMEN-PRO uses the rotation algorithm to fine-tune the architecture for higher accuracy and resource efficiency, leveraging physical properties of optical systems. On MNIST family, LUMEN-PRO achieves up to $13.51\%$ higher accuracy and $4\times$ and $2\times$ better cost efficiency than single-task and sota MTL DONN methods. On CelebA, LUMEN-PRO improves accuracy by up to $49.58\%$ compared to SoTA MTL DONN algorithms. In energy efficiency, LUMEN-PRO achieves up to $10^5\times$ and $10\times$ speedup in throughput, with $5,969\times$ and $680\times$ energy efficiency gain compared with IBM TrueNorth and nanophotonic, respectively. LUMEN-PRO enables flexible adjustment and identification of suitable models across datasets, facilitating efficient search for energy-efficient MTL DONN.

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
