# OpenReview forum: "LUMEN-PRO: Automating Multi-Task Learning on Optical Neural Networks with Weight Sharing and Physical Rotation"
_ICLR.cc/2024/Conference — Submitted to ICLR 2024_

### Official Review · Reviewer_njPU · 2023-10-26

**Soundness:** 3 good
**Presentation:** 3 good
**Contribution:** 3 good
**Rating:** 8
**Confidence:** 3

**Summary:**

This paper proposes an automated multi-task learning (MTL) framework dubbed LUMEN-PRO dedicated to diffractive optical neural networks (DONN). Then, the authors leverage the rotatability of the physical system and replace task-specific layers with the rotation of the corresponding shared layers. Both effectively reduce the memory footprint.

Experiments also show that the proposed LUMEN-PRO provides up to 49.58% higher accuracy and 4x better cost efficiency than single task and prior art methods.

**Strengths:**

1. The paper organization is great. Even though I do not have a relevant background on DONN, I can still follow the logic to understand the paper. E.g., Table 1 provides a good summary of current MTL methods and how the proposed one is better or more comprehensive.

2. The proposed method leverages the rotatability of the physical system to fine-tune the multi-task DONN. It is like the spatial shift to CNNs and helps with the generalization ability learning of such models.

3. Experiments show that the proposed methods achieve better task accuracy and cost efficiency than previous methods.

**Weaknesses:**

I am not an expert on DONN. As for the MTL and NAS:

The idea sounds like a combination of NAS and MTL. What is unique here for DONN? Is this method the general method that can be applied to other CNN or Transformer models?

You mentioned that the rotation mechanism has a physical meaning, what is that? Why is the rotation different from spatial shifts in CNNs?

As for the experiments, MNIST and CelebA are relatively small datasets, why do you consider larger ones? Is that because such DONN has some generalization or scalability issue preventing it from adapting to large scales?

**Questions:**

See Weaknesses

---

> ### Author Response · Authors · 2023-11-18
> **Thank you very much for your recognition of our work! Please find our responses to your comments below.**
>
> Your time in evaluating our work is sincerely appreciated. We would like to address your concerns and questions as below.
>
> ***Q1: What is unique for DONN? Is this method the general method that can be applied to other CNN or Transformer models?***
>
> R1: Thanks for your question and concern regarding the practical usage of our designed algorithm. We would address your questions in two folds.
>
> - *Unique for DONN.*
>
> In addition to the AutoMTL method, we also want to emphasize the significance of the rotation of the physical layer for the DONN system. While AutoMTL enables the discovery of an efficient multi-task framework, the rotation of the physical layer further facilitates a reduction in memory storage, allowing it to match that of a single-task model. This achievement establishes a memory lower bound, making the rotation of the physical layer an integral and noteworthy aspect.
>
> When doing inference on the physical optical device, the "rotation" action takes place after completing the inference for one task, facilitating the system's transition to conducting inference on another task. Due to the rotational capability of the physical system, there is no need to disassemble and reassemble the entire system or replace all the diffractive layers when switching tasks. Rather, we can simply rotate the corresponding layer or layers based on the emulated MTL model. This rotational approach offers cost savings by eliminating the need for additional printing materials and reduces the time required for hardware reconfiguration.
>
> - *Generalization of LUMEN-PRO.*
>
> The AutoMTL method is applicable to any CNN and Transformer model with ease. The physics-aware weight sharing rotation algorithm in our LUMEN-PRO is specifically designed for DONN systems due to the non-reconfigurable feature of the optical hardware system. However, we believe that with certain adjustments, this algorithm can also be applied to any model that has a square-shaped feature map.
>
> &nbsp;
>
> ***Q2: You mentioned that the rotation mechanism has a physical meaning, what is that? Why is the rotation different from spatial shifts in CNNs?***
>
> R2: “Physical meaning” meaning that when switching to different tasks, we can simply rotate the corresponding layer or layers based on the searched multi-task configurations.  There is no need to disassemble and reassemble the entire system or replace all the diffractive layers when switching. In practical deployment of DONN systems, the trained weights in the model (diffractive layers) are fabricated and deployed for all-optical inference, which is non-reconfigurable. Thus, to further reduce the system cost for multiple tasks learning with the deployed DONN system, we proposed the physical rotation mechanism for weights sharing where the cost efficiency is improved by 4x when dealing with 4 tasks compared with single task systems (Table 2). More importantly, the rotation mechanism works for symmetric but non-configurable physical systems.
>
> The rotation of diffractive layers and spatial shifts in CNNs are two distinct concepts used in different contexts.  Spatial shifts of CNNs refer to the translation or shifting of input data within the network. It involves moving or shifting the positions of input images or feature maps in the horizontal and vertical directions. Spatial shifts are typically used to achieve spatial invariance, allowing the network to recognize objects or patterns regardless of their specific position in the input image. In summary, the rotation of diffractive layers pertains to modifying the physical properties of the diffractive optics itself, whereas spatial shifts in CNNs involve shifting or translating input data within the network to achieve spatial invariance and robustness to translation.

---

> > ### Author Response · Authors · 2023-11-18
> > **Continue of Rebuttal**
> >
> > ***Q3: Why do you consider larger dataset? Is that because DONN has generalization or scalability issues preventing it from adapting to large scales?***
> >
> > R3: The current DONN system is efficient and good at dealing with large-size and large-scale datasets. For both dataset, we expand the input images into 200x200 for training and implementations by numerical interpolation. The light signal in the DONN system is cheap and easy to expand its diameter for larger dimensions of the inputs for scalability.
> > However, the system can be limited in dealing with complex datasets, such as RGB datasets, information-rich datasets. For example, RGB image classification can be handled by utilizing ONNs with multiple light channels in order to process large-scale R,G, and B information, but the accuracy is pretty far from conventional NNs for now (e.g., ImageNet). Another more complex dataset handled by the DONN system is the moving MNIST/augmented MINST. For example, in Figure 3 of [1], which demonstrates that such multi-variants objectives can be successfully classified. Recently some group demonstrate the dynamic version of it, see https://static-content.springer.com/esm/art%3A10.1038%2Fs41566-021-00796-w/MediaObjects/41566_2021_796_MOESM2_ESM.mp4
> >
> > Additionally, DONN can extend its capabilities to process graph-structured data. We are investigating the potential of the monolithic free-space DONN graph learning system for graph-based deep learning. Our initial experimental results utilized standard graph learning datasets Cora and Citeseer, demonstrating accuracies of 83.7% and 74.6%, respectively. [2] also merges diffractive photonic computing units (DPUs) and on-chip optical devices to execute semisupervised node classification on large-scale graph datasets like Cora, Citeseer, and the Amazon photo datasets. Furthermore, it is capable of performing graph classification for skeleton-based human action recognition.
> >
> > &nbsp;
> >
> > [1] Mengu, Deniz, Yair Rivenson, and Aydogan Ozcan. "Scale-, shift-, and rotation-invariant diffractive optical networks." ACS Photonics 8.1 (2020): 324-334.
> >
> > [2] Yan, T., Yang, R., Zheng, Z., Lin, X., Xiong, H., & Dai, Q. (2022). All-optical graph representation learning using integrated diffractive photonic computing units. Science Advances, 8(24), eabn7630.

---

> > > ### Comment · Reviewer_njPU · 2023-11-23
> > > **Response to author rebuttal**
> > >
> > > After carefully reviewing the author's responses, I find that my concerns have been addressed. The innovative rotation mechanism in the fabricated optical system adds a valuable dimension, rendering it "reconfigurable" and capable of handling diverse tasks. The research of optical neural networks is a relatively new contribution to the neural network community, and the identified properties appear to complement the existing DONN system effectively. Consequently, I am inclined to maintain my positive rating for this work.

---

### Official Review · Reviewer_nw67 · 2023-10-28

**Soundness:** 3 good
**Presentation:** 2 fair
**Contribution:** 2 fair
**Rating:** 6
**Confidence:** 4

**Summary:**

This paper proposes a multi-task learning optical neural network framework, LUMEN-PRO, which uses the physical principles to effectively improve the performance and cost of the model for multi-tasks.

**Strengths:**

1. The idea of modeling multitasking by rotating the physical layer is novel and interesting and can effectively reduce costs.

2. Diffractive optical neural networks-based methods can greatly improve the inference performance.

3. This area of research is rare and can increase the diversity of the ML community.

**Weaknesses:**

1. The method depends on the rotation of the physical layer. However, the physical layer has at most four directions. Therefore, the method only supports most four tasks.

2. The method is mainly derived from the AutoMTL [1] method.

3. The presentation is not clear enough. Some details are not included in the paper. This can be seen in the questions.

4. The Figures are not annotated; thus, it is difficult to understand the method directly by looking at them.

[1] Zhang, Lijun, Xiao Liu, and Hui Guan. "Automtl: A programming framework for automating efficient multi-task learning." Advances in Neural Information Processing Systems 35 (2022): 34216-34228.

**Questions:**

1. What is the meaning of the  LUMEN-PRO in Figure 5? As I understand, this network can only make the inference function different under different tasks by rotating the layers.

2. Why the LUMEN-PRO’s performance can exceed the single task in Figure 5? Normally, it works best to use a separate model for each task.

---

> ### Author Response · Authors · 2023-11-18
> **Thank you for your valuable feedbacks and suggestions. Please find our responses to your comments below.**
>
> We sincerely appreciate the time you've taken to evaluate our work. We would like to address your concerns and questions as below.
>
> ***Q1: The method only supports most four tasks.***
>
> R1: While it is true that the physical layer typically has four directions, the method does not necessarily limit the number of tasks to only four. Firstly, not all tasks are necessarily to choose the task-specific option simultaneously at a specific layer. Furthermore, during training, we allocate more weights to the "sharing" option compared to the "task-specific" option to encourage sharing operators across tasks. Secondly, even if all tasks select the task-specific option at the same layer, resulting in some of the tasks sharing the same rotation angle and therefore having identical weights at that layer, these tasks would still exhibit distinct operator selection or rotation angles in other layers, which makes them still have different model structures and weights. The rotation of the physical layer used in our LUMEN-PRO enables efficient sharing of weights among tasks, allowing for a larger number of tasks to be supported and also greatly reduces the memory overhead.
>
> &nbsp;
>
> ***Q2: The method is mainly derived from the AutoMTL method.***
>
> R2: Thank you for highlighting this important aspect. **We emphasize the significance of the rotation of the physical layer in the DONN system**. The rotation of the physical layer further facilitates a reduction in memory storage, allowing it to match that of a single-task model. This achievement establishes a memory lower bound, making the rotation of the physical layer an integral and noteworthy aspect.
>
> When performing inference on the physical optical device, the "rotation" action takes place after completing the inference for one task, facilitating the system's transition to conducting inference on another task. Due to the rotational capability of the physical system, there is no need to disassemble and reassemble the entire system or replace all the diffractive layers when switching tasks. We can simply rotate the corresponding layer or layers based on the searched multi-task configurations. This rotational approach offers cost savings by eliminating the need for additional printing materials and reduces the time required for hardware reconfiguration.
>
> &nbsp;
>
> ***Q3: The Figures are not annotated.***
>
> R3: We greatly appreciate your valuable suggestions. We will incorporate your comments into the paper and include them in the final version.
>
> &nbsp;
>
> ***Q4: What is the meaning of the LUMEN-PRO in Figure 5? As I understand, this network can only make the inference function different under different tasks by rotating the layers.***
>
> R4: Yes, your understanding is correct. This network's ability to alter the inference function for distinct tasks is achieved by rotating the layers. In Figure 5, we compare LUMEN-PRO w/o rotation and LUMEN-PRO w/ rotation with other state-of-the-art (sota) multi-task learning (MTL) models. LUMEN-PRO w/o rotation is an MTL DONN obtained using the AutoMTL algorithm, with each task-specific layer having its specific weight. LUMEN-PRO w/ rotation refers to an MTL DONN obtained by combining AutoMTL with rotation design. This model achieves the same memory footprint as a single-task model since all task-specific layers are derived by physically rotating the shared layers. The purpose of comparing the accuracy between these models is to conduct an ablation study highlighting two points: (1) the effectiveness of the MTL DONN obtained through AutoMTL, which outperforms both single-task models and other SoTA MTL models in terms of accuracy, and (2) the effectiveness of the added rotation algorithm in LUMEN-PRO. This algorithm not only surpasses the accuracy of both single-task models and other sota MTL models but also achieves a memory lower bound with minimal accuracy sacrifice compared to not incorporating the rotation algorithm.
>
> &nbsp;
>
> ***Q5: Why the LUMEN-PRO’s performance can exceed the single task in Figure 5?***
>
> R5: Multi-task learning models can outperform single-task models due to several reasons. Firstly, the MTL model can improve generalization by learning shared representations and identifying common patterns across multiple related tasks. Secondly, multi-task learning enables the model to leverage limited data more efficiently by using information from multiple tasks, resulting in better performance overall. Thirdly, it acts as a form of regularization, preventing overfitting and enhancing the model's ability to generalize. Lastly, by sharing weights, the memory footprint of multi-task learning model is significantly reduced for all tasks compared to having a model for each task. The effectiveness of multi-task learning models depends on task relatedness.

---

### Official Review · Reviewer_edBq · 2023-10-30

**Soundness:** 2 fair
**Presentation:** 2 fair
**Contribution:** 2 fair
**Rating:** 3
**Confidence:** 4

**Summary:**

This work describes a multi-task learning approach for a specific optical neural network named Diffractive Optical Neural Networks (DONN). It leverages the rotability of the physical system to share the same module across different tasks. Experiments was conducted on MNIST and its variants, and a face attribute dataset. The proposed method is able to outperform existing DONN multi-task learning method on accuracy with lower cost to fabricate the system.

**Strengths:**

-	The idea of enabling layer sharing in a optical neural network is interesting. The authors use some existing gradient-based architecture search algorithm (Automtl Neurips 22) and adapt it to the DONN scenario.
-	The proposed method achieves significant performance gain on MNIST and Celeb-Faces dataset compared to existing multi-task methods such as VanillaMT and RubikONN.

**Weaknesses:**

-	The application of designing multi-task learning method for the DONN is too narrow. DONN is just one type of optical neural network and there is no evidence that this approach generalizes to other physical neural networks. The method may not have much practical usages in real life.
-	The experiments conducted seems only from a mathematical perspective. If we put this solution to produce physical systems, will there be accuracy degeneration due to imperfect fabrication? And is the proposed rotation-based sharing method practical in real fabricated system? The authors did not address these issues.
-	Due to my lack of experience with this field, I do not understand a lot of technical details in this work. I believe the authors can improve on the explanation of the key concept to make it easy to understand. For example, the Figure 3 is really confusing. What does a node mean? What does the numbers in the node blocks mean? Are they network weights? There are a bunch of switches on the figure. What is the functionality of these switches and how do they work? Another key concept is the rotation-based layer sharing. What exactly does rotation mean in this scenario? How does such rotation facilitate weight sharing?

**Questions:**

-	Table 3 is kind of confusing. It seems to contain both ASIC-based solution and physical neural networks. How do you measure the throughput of an optical neural network? The proposed framework has very high throughput but is it really possible in a real system? Since you need to switch the input image physically at such a fast rate. And what does ``Accuracy’’ mean in this table? Is it just the testing accuracy on MNIST?

---

> ### Author Response · Authors · 2023-11-18
> **Thank you for your feedback. Please see our responses below and hope that addresses your concerns.**
>
> We sincerely appreciate the time you've taken to evaluate our work. We would like to address your concerns and questions as below.
>
> ***Q1: The application of designing multi-task learning method is too narrow and no practical usages.***
>
> R1: To address your concern regarding the practical application of our designed algorithm, let us explain in two folds.
>
> - *Practical usages of multi-task learning on other physical neural networks.*
>
> Firstly, the proposed multi-task learning method has the potential to generalize to other physics-based neural networks, which utilizes nature physics as neural operators and are physically manufactured. For example, Figure 2 in [1] shows a physics-based neural network using broadband optical second-harmonic generation (SHG), and a mechanical version. The proposed methods can be extended beyond the targeted tasks (image classification), as long as the shared physical implementation can be realized, e.g., metasurface based imaging processing unit [2]. Note that in these systems (and ours), once the neural networks are fabricated, the rotation-based weight sharing will be able to share physical “weight” parameters without re-fabricate the system. For example, the optical and mechanical physics neural networks demonstrated in [1] can leverage completely our proposed methods in order to share the physically fabricated layers. Causally, this is a very similar idea to “Lenticular printing” in real life. At this point, we kindly request the reviewer to kindly consider the contributions of this paper fairly from the physical neural network point of view, instead of comparing it to conventional digital neural networks.
>
> Specifically for DONN, the rotation mechanism enables a physics-aware weights sharing. In practical deployment, the trained weights in the diffractive layers are fabricated and deployed for all-optical inference, which is non-reconfigurable. To further reduce the system cost for multiple tasks learning, we propose the physical rotation mechanism for weights sharing where the cost efficiency is improved by 4x when dealing with 4 tasks compared with single task systems (Table 2 in our paper). More importantly, the rotation mechanism works for symmetric but non-configurable physical systems.
>
> - *Practical usages of DONN.*
>
> Secondly, Free-space Diffractive Optical Neural Networks have practical applications across various domains [3][4]. They excel in high-speed computing tasks, enabling rapid processing for real-time image and video analytics, object recognition, and data classification. In optical image processing,  DONNs prove valuable for applications like image denoising, segmentation, and reconstruction[5][6]. They also excel in optical pattern recognition tasks, facilitating real-time recognition of complex patterns, document classification, biometric identification, surveillance systems, and optical character recognition[7][8]. DONNs contribute to neuromorphic computing systems, enabling machine learning, cognitive processing, and brain-inspired computing. Furthermore, they offer advantages in optical signal processing and sensing, enhancing Fourier analysis, beam shaping, spectral estimation, and enabling improved optical sensing capabilities. With their unique optical properties and parallel computing capabilities, DONNs present promising opportunities in the fields of research, industry, and technology. In this case, further exploration of DONNs applications, including multi-task learning, holds significant promise for addressing real-world challenges and achieving breakthroughs in various fields.
>
>
> [1] Wright, Logan G., Tatsuhiro Onodera, Martin M. Stein, Tianyu Wang, Darren T. Schachter, Zoey Hu, and Peter L. McMahon. "Deep physical neural networks trained with backpropagation." Nature 601, no. 7894 (2022): 549-555.
>
> [2] Li, L., Zhao, H., Liu, C., Li, L., & Cui, T. J. (2022). Intelligent metasurfaces: control, communication and computing. Elight, 2(1), 7.
>
> [3] Lin, X., Rivenson, Y., Yardimci, N. T., Veli, M., Luo, Y., Jarrahi, M., & Ozcan, A. (2018). All-optical machine learning using diffractive deep neural networks. Science, 361(6406), 1004-1008.
>
> [4] Sui, X., Wu, Q., Liu, J., Chen, Q., & Gu, G. (2020). A review of optical neural networks. IEEE Access, 8, 70773-70783.
>
> [5] Luo, Y., et.al. (2019). Design of task-specific optical systems using broadband diffractive neural networks. Light: Science & Applications, 8(1), 112.
>
> [6] Mengu, D., Veli, M., Rivenson, Y., & Ozcan, A. (2022). Classification and reconstruction of spatially overlapping phase images using diffractive optical networks. Scientific Reports, 12(1), 8446.
>
> [7] Liu, J., et.al. (2021). Research progress in optical neural networks: theory, applications and developments. PhotoniX, 2(1), 1-39.
>
> [8] Chen, R., Tang, Y., Ma, J., & Gao, W. (2023). Scientific Computing with Diffractive Optical Neural Networks. arXiv preprint arXiv:2302.10905.

---

> > ### Author Response · Authors · 2023-11-18
> > **Continue of Rebuttal**
> >
> > ***Q2: Will there be accuracy degeneration due to imperfect fabrication? Is the proposed method practical in real fabricated system?***
> >
> > R2: There are several approaches to deal with the accuracy degradation after physical deployment: (1) As shown in [2], we can use adaptive training to fine-tune the model regarding the optical devices. (2) We can integrate physics-aware codesign algorithms in [1] in our numerical emulation and training process, where the algorithm takes the imperfect fabrications into the training process for better hardware-software correlations. [1] provide the physical prototype for physical deployment, which shows the impressive correlation between numerical emulation and physical measurements.
> >
> > As for the rotation-based sharing method, it is practical after the fabrication system. As shown in Figure 2 in our paper, the diffractive layers (i.e., the weight matrix) are fabricated as 3D printed phase masks with square shape. To perform the rotation method, for example, in the system in Figure 2 of [3], the fabricated layers are inserted in slots. We can take the phase mask out of the slot, rotate an angle, and insert the layer back to the slot (these movements can be accomplished by a fast robotic arm designed), which is straightforward and realizes weights sharing at no extra fabrication cost for multiple tasks in DONN systems.
> >
> >
> > [1] Y. Li, R. Chen, W. Gao, and C. Yu. Physics-aware differentiable discrete codesign for diffractive optical neural networks. In Proceedings of the 41st IEEE/ACM International Conference on Computer-Aided Design, pp. 1–9, 2022.
> >
> > [2] T. Zhou, X. Lin, J. Wu, Y. Chen, H. Xie, Y. Li, J. Fan, H. Wu, L. Fang, and Q. Dai, “Large-scale neuromorphic optoelectronic computing with a reconfigurable diffractive processing unit,” Nature Photonics, vol. 15, no. 5, pp. 367–373, 2021.
> >
> > [3] X. Lin, Y. Rivenson, N. Yardimci, M. Veli, Y. Luo, M. Jarrahi, and A. Ozcan. All-optical machine learning using diffractive deep neural networks. Science, 361(6406): 1004–1008, 2018.

---

> > ### Comment · Reviewer_edBq · 2023-11-22
> > **Thanks for the comments**
> >
> > Thanks for the detailed comments. As requested by the authors, I carefully reconsider the contributions of this paper from the physical neural network point of view. However, I still do not find this work to have sufficient contribution in this direction. The proposed methods are heavily borrowed from NAS literature and the only adaptation the authors did is to add the "rotation-based sharing" mechanism into it. The rotation-based sharing is also proposed by previous work and is not original by this work. Further, the authors claim that the proposed method can be extended to other physics-based neural networks, without providing any experimental verifications. I believe more experiments on other type of physics-based neural networks is needed to make this claim valid. With all those updated considerations, I still holding my original recommendation of reject.

---

> > > ### Author Response · Authors · 2023-11-22
> > > **Reply about the concerns.**
> > >
> > > Thank you very much for the reply. We would like to address your concerns as below.
> > >
> > > Firstly, the primary focus of our paper is to extend DONN into the realm of multi-task learning (MTL), aiming to achieve the most efficient and lightweight MTL DONN with higher accuracy. Our framework significantly differs from AutoMTL, as it is specifically redesigned for the DONN system and merged with the rotation algorithm for the purpose of automated search and rotation for better performance. Our experimental results demonstrate a memory lower bound (cost-efficient MTL system with the hardware cost) as low as that of the single-task model, while achieving up to a 50% improvement in accuracy (Figure 7 for CelebA dataset), a milestone not reached by previous state-of-the-art approaches. We emphasize the importance of understanding why and how our pipeline incorporates existing techniques.
> > >
> > > Secondly, we think your comment/suggestion about reproducing and demonstrating the results of NATURE papers (e.g., the one mentioned in our earlier response ([1])) for a system/algorithm in ICLR submission is not fair. If you have any pertinent papers related to PNN that could serve for comparison or alignment with our algorithm algorithmically, we would appreciate it if you could provide the relevant references.
> > >
> > > Thirdly, in the domain of NAS, many papers investigate diverse combinations of techniques to improve performance. For example, [2] adopt the traditional automatic search algorithm onto the transformer model to enable NAS on transformer-based architectures. [3] modified the search space of the previous algorithm and achieve better performance. Thus, we believe our work possesses its novelty in exploring the MTL algorithms in DONN systems. We sincerely hope for a reconsideration of the score.
> > >
> > > [1] Wright, Logan G., Tatsuhiro Onodera, Martin M. Stein, Tianyu Wang, Darren T. Schachter, Zoey Hu, and Peter L. McMahon. "Deep physical neural networks trained with backpropagation." Nature 601, no. 7894 (2022): 549-555.
> > >
> > > [2] Chen, M., Peng, H., Fu, J., & Ling, H. (2021). Autoformer: Searching transformers for visual recognition. In Proceedings of the IEEE/CVF international conference on computer vision (pp. 12270-12280).
> > >
> > > [3] Chen, M., Wu, K., Ni, B., Peng, H., Liu, B., Fu, J., ... & Ling, H. (2021). Searching the search space of vision transformer. Advances in Neural Information Processing Systems, 34, 8714-8726.

---

> ### Author Response · Authors · 2023-11-18
> **Continue of Rebuttal**
>
> ***Q3: Confusing about Figure 3.***
>
> R3: Thank you for your questions. We will address your questions in two parts.
>
> - ***What does a node mean? What does the numbers in the node blocks mean? What is the functionality of these switches?***
>
> The input DONN backbone model follows a single-task architecture, with numbers representing the weights in each layer. In Figure 3, we utilize a three-layer DONN as an example, where each layer is represented by a 2x2 matrix. To represent the DONN backbone model as a computation graph, we employ nodes to symbolize the layers. In this context, the notation $node_i$ is used to refer to layer $i$ within the backbone model.
>
> In order to transition the single-task backbone into a multi-task model, we enhance each node as a computation unit ($CU$). If there are $N$ tasks in the design, each $CU$ would consist of $N$ blocks, with each block comprising three operators: (1) the basic shared operator ($bsc$), (2) a task-specific copy of the basic shared operator ($spc^t$), or (3) a skip operator ($skp^t$). The basic shared operator ($bsc$) is common and shared across all $N$ blocks (tasks). All these *CUs* form our multi-task supermodel.
>
> Our objective is to learn the policy for each task, which involves determining the suitable operator to select for each task in every layer within the *CUs*. The policy is denoted by "switches" (*P*), such as $P_2^{t_1}$, representing the policy for task 1 in layer 2. After the entire training process, $P_2^{t_1}$ can take on one of three possibilities: (1,0,0) represents the selection of the shared operator (share the same weight with other tasks) in layer 2 for task 1; (0,1,0) represents the selection of the task-specific operator (with its own specific weights in that layer) for task 1 in layer 2, and (0,0,1) indicates the decision to skip that layer for task 1.
>
> - ***What does rotation mean? How does rotation facilitate weight sharing?***
>
> We present the detail of "rotation" in *Section 4.2 Rotation Algorithm*. In this context, "rotation" refers to obtaining the task-specific layer through a 90-degree / 180-degree / 270-degree rotation of the shared layer. For example, in Figure 4 (a), if we denote layer 1 of task 1 as $L_1^1$, the weights in the first layer of task 2 ($L_2^1$) and task 3 ($L_3^1$) are derived from a 90-degree and 180-degree clockwise rotation, respectively, of the weights in $L_1^1$. In the second layer, both task 1 and task 2 select the shared operator, resulting in them sharing the same weight in this layer ($L_1^2 = L_2^2$), while task 3 chooses the task-specific operator. As a result, the weights in the second layer of task 3 are obtained through a 90-degree rotation of $L_1^2$. In model training, the degrees of rotation are actually randomly assigned.
>
> By leveraging "rotation," it is possible to construct a multi-task learning model with equivalent parameter size and memory storage as a single-task model. This is achieved by having all tasks within each layer either share the same weights or acquire their weights through rotation from the weights of other tasks. Moreover, when deploying the emulated model onto a physical optical device, such as a 5-layer multi-task model supporting 4 tasks, only 5 phase masks need to be fabricated. By rotating some of these masks, the inference process can support all four tasks efficiently.

---

> > ### Author Response · Authors · 2023-11-18
> > **Continue of Rebuttal**
> >
> > ***Q4: Confusing about Table 3.***
> >
> > R4: Thank you for your questions. We will address your questions separately.
> >
> > - ***How do you measure the throughput of an optical neural network?***
> >
> > Table 3 showcases the theoretical throughput of the free-space DONN. The presented data is comparable to the theoretical throughput figures shared by HolyLight paper, which encompass the performance of ISAAC and DaDianNao (simulated results with theoretical bounds). We follow the same setting. The distinct characteristic of the free-space DONN lies in its ability to utilize all-optical processes. Leveraging the inherent properties of light, including natural light diffraction and swift light signal phase modulation at the speed of light, the free-space DONN exhibits the (theoretically) potential to achieve a throughput of $10^5$.
> >
> > - ***The high-throughput framework's practical feasibility raises concerns due to the physical input switching required.***
> >
> > The "rotation" action does not occur during inference of each individual task. Instead, it takes place after completing inference for one task, facilitating the system's transition to conducting inference on another task. Due to the rotational capability of the physical system, **there is no need to disassemble and reassemble the entire system or replace all the diffractive layers when switching tasks**. Rather, we can simply rotate the corresponding layer or layers based on the searched multi-task configurations. This rotational approach offers cost savings by eliminating the need for additional printing materials and reduces the time required for hardware reconfiguration. It's important to note that the discussion of "cost" and "time" in this context is not reflected in Table 3.
> >
> > - ***What does "Accuracy’’ mean in Table 3?***
> >
> > As mentioned in the "Energy Efficiency Comparison" paragraph under *Section 5.2.2 Accuracy-Cost Comparison*, Table 3 presents the accuracy results specifically for MNIST. For LUMEN-PRO, this implies that we have already assembled the physical diffractive layers and related hardware components of the DONN. The measurements provided in Table 3 focus solely on the throughput and power requirements associated with MNIST.

---

### Comment · Area_Chair_ytzV · 2023-11-19
**Please engage in reviewer-author discussion**

Dear reviewers,

The paper got diverging scores. The authors have provided their response to the comments. Could you look through their response and other reviews and engage into the discussion with authors? See if their response changes your assessment of the submission?

Thanks!
AC

---

### Author Response · Authors · 2023-11-22
**Additional Responses**

Dear Reviewers,

&nbsp;

As today marks the final day of the discussion period, we wanted to check in and see if any lingering concerns or questions remain following our last response.

We are fully committed to addressing any further questions or concerns you might have. Additionally, we respectfully request that you consider re-evaluating our manuscript, and we would be grateful for a reconsideration of the score, accompanied by your justifications for any adjustments.

We eagerly anticipate engaging in open discussions and are keen to benefit from your expertise and guidance.

&nbsp;

Sincerely,

The authors

---

### Meta-Review · Area_Chair_ytzV · 2023-12-07

**Metareview:**

This work presents a novel approach to multi-task learning in optical neural networks, leveraging physical rotation for weight sharing. It can bring some improvement in task accuracy and cost efficiency.

Strengths:
Most reviewers agree on the novelty of  rotating the physical layer in DONN and its impressive performance.

Weaknesses:
The main concerns from reviewers lie on the its practicality, generalizability, and clarity in presentation.  In addition the novelty of this method is limited in its similarity with existing NAS literature.

Taking all factors into account, the AC does not recommend this paper to be accepted at ICLR because of the limited novelty, experimental validation, practicality and generalizability, and clarity in presentation.

**Justification For Why Not Higher Score:**

This work should improve on explaning the practical usability, generalization to other type of PNNs of the proposed method, and clarity of presentation. The similarity with NAS downweighs its novelty and experimental validation should be further improved.

**Justification For Why Not Lower Score:**

This work is recommended to be rejected.

---

### Decision · Program_Chairs · 2024-01-16

Reject